# PPAR Alpha Activation by Clofibrate Alleviates Ischemia/Reperfusion Injury in Metabolic Syndrome Rats by Decreasing Cardiac Inflammation and Remodeling and by Regulating the Atrial Natriuretic Peptide Compensatory Response

**DOI:** 10.3390/ijms24065321

**Published:** 2023-03-10

**Authors:** María Sánchez-Aguilar, Luz Ibarra-Lara, Agustina Cano-Martínez, Elizabeth Soria-Castro, Vicente Castrejón-Téllez, Natalia Pavón, Citlalli Osorio-Yáñez, Eulises Díaz-Díaz, María Esther Rubio-Ruíz

**Affiliations:** 1Department of Pharmacology, Juan Badiano 1, Sección XVI, Tlalpan, Mexico City 14080, Mexico; msanchezaguilar@gmail.com (M.S.-A.); luzibarralara@gmail.com (L.I.-L.); pavitonat@yahoo.com.mx (N.P.); 2Department of Physiology, Instituto Nacional de Cardiología Ignacio Chávez, Juan Badiano 1, Sección XVI, Tlalpan, México City 14080, Mexico; agustina.cano@cardiologia.org.mx (A.C.-M.); vicente.castrejon@cardiologia.org.mx (V.C.-T.); 3Department of Cardiovascular Biomedicine, Juan Badiano 1, Sección XVI, Tlalpan, México City 14080, Mexico; elizabeth.soria@cardiologia.org.mx; 4Departamento de Medicina Genómica y Toxicología Ambiental, Instituto de Investigaciones Biomédicas, Universidad Nacional Autónoma de México, Ciudad Universitaria, Apartado Postal 70228, Ciudad de México 04510, Mexico; citlalli.osorio@iibiomedicas.unam.mx; 5Laboratorio de Fisiología Cardiovascular y Transplante Renal, Unidad de Investigación UNAM-INCICH, Instituto Nacional de Cardiología Ignacio Chávez, Juan Badiano 1, Sección XVI, Tlalpan, México City 14080, Mexico; 6Department of Reproductive Biology, Instituto Nacional de Ciencias Médicas y de la Nutrición “Salvador Zubirán”, Vasco de Quiroga 15, Sección XVI, Tlalpan, México City 14000, Mexico; eulisesd@yahoo.com

**Keywords:** ischemia/reperfusion injury, natriuretic peptides, metabolic syndrome, cardiac remodeling, cardiac inflammation, PPAR alpha agonist

## Abstract

Metabolic syndrome (MetS) is a cluster of factors that increase the risk of developing diabetes, stroke, and heart failure. The pathophysiology of injury by ischemia/reperfusion (I/R) is highly complex and the inflammatory condition plays an important role by increasing matrix remodeling and cardiac apoptosis. Natriuretic peptides (NPs) are cardiac hormones with numerous beneficial effects mainly mediated by a cell surface receptor named atrial natriuretic peptide receptor (ANPr). Although NPs are powerful clinical markers of cardiac failure, their role in I/R is still controversial. Peroxisome proliferator-activated receptor α agonists exert cardiovascular therapeutic actions; however, their effect on the NPs’ signaling pathway has not been extensively studied. Our study provides important insight into the regulation of both ANP and ANPr in the hearts of MetS rats and their association with the inflammatory conditions caused by damage from I/R. Moreover, we show that pre-treatment with clofibrate was able to decrease the inflammatory response that, in turn, decreases myocardial fibrosis, the expression of metalloprotease 2 and apoptosis. Treatment with clofibrate is also associated with a decrease in ANP and ANPr expression.

## 1. Introduction

Metabolic syndrome (MetS) is an entity characterized by various risk factors, such as hypertension, obesity, insulin resistance, and atherogenic dyslipidemia that includes reduced high density lipoprotein (HDL) cholesterol and increased triglycerides [1]. This pathology is associated with a sedentary lifestyle and increased caloric intake as well as genetic factors [2]. The association of MetS and its components with increased risk of adverse cardiovascular outcomes, morbidity, and mortality is well established. There are previous reports that MetS doubles the incidence of coronary artery disease, increases the progression of atheromatous plaque, and elevates the incidence of cardiac failure [3].

Heart failure remains the leading cause of death, morbidity, and medical expenses worldwide. Myocardial infarction and the pathophysiology of injury by ischemia/reperfusion (I/R) are highly complex. The production of several cytokines increases in myocardial damage by I/R, including tumor necrosis factor alpha (TNF-α), interleukin-6 (IL-6), interleukin-1 beta (IL-1β), and transforming growth factor-beta (TFG-β1) [4,5]. Furthermore, inflammation plays an important role in cardiac remodeling by regulating three processes: (1) increasing deposition of types I and III collagen and extracellular matrix crosslinking (fibrosis), (2) increasing the expression and activity of metalloproteinases (MMPs), such as MMP-2 and MMP-9, and (3) stimulating infiltration of leukocytes, which increases the inflammatory response. Altogether, these processes lead to matrix remodeling and cardiomyocyte apoptosis that may contribute to the development and progression of heart failure [6,7].

The myocardial injury that occurs following a period of I/R is not yet completely understood in its underlying pathophysiological mechanism. In response to myocardial damage, a cascade of compensatory events occurs that moderates the cardiac remodeling process including kinins, urocortins, adrenomedullin, incretins, and natriuretic peptides (NPs) [8]. NPs form part of a family of humoral components that are involved in cardiovascular homeostasis but also have effects at the endocrine level and participate in energy balance, mitochondrial biogenesis, respiration, and lipid oxidation [9]. NPs have been proposed as therapeutic strategies for obesity due to their production by other organs, such as adipose tissue, liver, and skeletal muscle, among others, and in diseases, such as MetS and type 2 diabetes [10]. There are three types of NPs: atrial natriuretic peptide (ANP), brain natriuretic peptide (BNP), and C natriuretic peptide (CNP), whose expression is regulated by several factors including the pro-inflammatory cytokines present in pathological conditions [11]. It is well known that ANP and N-terminal pro-BNP (NT-ProBNP) are powerful clinical markers of heart failure; however, it is unclear whether enhancement of ANP and BNP expression is protective or if it aggravates myocardial damage.

The activities of NPs are mediated by natriuretic peptide receptors A, B, and C, which are cell surface, single-span transmembrane receptors linked to the activity of an intrinsic guanylate cyclase. The three NP receptors differ in structure, biological effects, and ligand specificity; type A and B receptors activate downstream signaling pathways through the synthesis of the intracellular second messenger, cGMP, while type C receptor is mainly involved in their clearance. Receptor type A has greater affinity for ANP but can also bind to BNP [12,13]. The expression of the NPs’ receptors is regulated by several factors, such as vitamin D, angiotensin II, and endothelin, and by transcription factors, such as cGMP response element-binding protein (CREBP). Patients with hypertension, obesity, or MetS have lower levels of ANP, NT-proBNP, and ANP receptor [11,14,15].

On the other hand, several therapeutic approaches have been proposed for the control of MetS due to its multifactorial nature. Specifically, drugs, such as statins or fibrates, have been extensively used to treat dyslipidemia [1]. Moreover, some reports have shown the pleiotropic effects of fibrates to protect the heart from damage through the activation of peroxisome-proliferator-activated receptors (PPAR). Three members of the PPAR family, PPARα, PPARγ, and PPARβ/δ, have been investigated widely. PPARα plays important roles in many processes including inflammation, remodeling, metabolism, mitochondria biogenesis, and apoptosis [16]. We and others have reported that PPARα agonists constitute an effective treatment to decrease myocardial damage in several experimental models. The anti-inflammatory effect of PPARα in myocardial injury is mainly through inhibition of pro-inflammatory signaling pathways and improvement of the lipid profile. Moreover, PPARα also modulates the myocardial renin–angiotensin pathway and resets the insulin signaling pathway [17,18]. Although PPARα signaling is known to be associated with a cardioprotective effect, there is little evidence regarding the effect of PPARα on the regulation of NPs and their receptors in myocardial injury. In this work, we hypothesized that a PPARα agonist, clofibrate, exerts a cardioprotective effect by decreasing myocardial inflammation, remodeling, and apoptosis in a MetS rat model. Therefore, the aim of this study was to evaluate the effect of clofibrate on the expression of NPs and ANPr in hearts from MetS rats subjected to I/R injury.

## 2. Results

The body and serum biochemical parameters confirm the presence of MetS in our experimental rat models and are presented in Table 1. The MetS animals were hypertensive, dyslipidemic (high triglycerides and non-HDL cholesterol and low HDL cholesterol concentrations), and they had central obesity and insulin resistance. Clofibrate treatment decreased central obesity and body weight and improved lipid profile and insulin sensitivity in MetS rats. In contrast, only the levels of non-HDL-c were significantly decreased with the same treatment in control (Ct) rats (Table 1).

Figure 1 shows concentrations of IL-1β, IL-6, and TNF-α in left ventricles from Ct and MetS rats. Our results show that MetS sham-operated rats had significantly higher levels (approximately three times more) of these pro-inflammatory interleukins when compared to Ct-Sh animals. I/R conditions promoted an increase in the concentrations of these variables from Ct and MetS rats and the 7-day clofibrate pre-treatment prevented this increase in the same proportion in the experimental groups (approximately 45% for IL-1β, 30% for IL-6, and 55% for TNF-α).

Dying cardiomyocytes secrete cytokines to recruit leukocytes into the infarcted myocardium, and this intense inflammatory reaction initiates a reparative response that includes synthesis of collagen. For this reason, we decided to perform hematoxylin–eosin stain (HE) and picrosirius red (PSR) staining to evaluate the number of infiltrated cells and ventricular fibrosis in all experimental groups (Figure 2A,B). The hearts from rats with MetS presented more infiltrated cells and collagen deposition when compared to hearts from Ct animals (Figure 2A,B). HE staining of tissues revealed that I/R injury promoted cell swelling and the disruption and irregular arrangement of myocardial fibers when compared to sham groups; these changes are more evident in hearts from MetS rats (Figure 2A). We also observed numerous inflammatory cells infiltrated in the left ventricles of MetS rats after I/R injury (yellow arrows). When the MetS animals were pre-treated with clofibrate, these showed minor morphological changes and a decrease in the number of infiltrated cells (Figure 2A), which is an indication that some pro-inflammatory process has been disactivated. Sirius Red staining showed a significant increase in the collagen volume fraction (CVF) as deposition of collagen I and III fibers in the left ventricle from MetS rats in Sh-operated and under I/R conditions. Treatment with clofibrate was associated with the decrease of the CVF. There were no significant differences among the groups in control rats (Figure 2B).

Figure 3 shows the levels of protein expression of MMP-2 that participate in the cardiac remodeling process. Western blot analysis indicated that in sham-operated and under I/R conditions, hearts from MetS rats had a higher level of MMP-2 expression compared to the Ct group. However, clofibrate treatment was able to significantly decrease the MMP-2 expression in ventricles from both Ct and MetS animals compared to vehicle-treated corresponding groups.

The immunoexpression of ANP and ANP receptor in our experimental groups was determined to assess myocardial NPs’ response in I/R injury. The expression levels of ANP and ANPr were similar in hearts from Ct and MetS animals. There was a marked increase in the ANP/ANPr signal in the ventricles of MetS-V-IR rats compared with sham-operated MetS animals (Figure 4 and Figure 5). In contrast, MetS rats pre-treated with clofibrate displayed a significant decreased in the density of ANP and its receptor. There was no significant difference in the levels of expression of ANP and ANPr in ventricles from Ct groups.

The concentration of NT-proBNP was determined in the myocardia from Ct and MetS animals due to the fact that it is considered a biomarker for diagnosis and severity of heart failure. NT-proBNP concentrations were not statistically significantly different among groups (Figure 6).

The presence of ANP and its receptor in mitochondria was investigated using immunogold labeling for electronic microscopy because cardiac NPs have beneficial effects at the level of this organelle. Figure 7 shows that ANP and ANPr were present in greater quantities in mitochondria and fibers from ventricles from MetS-Sh rats than in hearts from Ct-Sh rats. Myocardial I/R injury in MetS rats was associated with intense immunoreactivity that was concentrated on the fibers and mitochondria. The MetS group treated with clofibrate showed a decrease in the expression of ANP and ANPr (Figure 7A,B). In contrast, the gold marks in ventricles from Ct groups remained unchanged.

The effect of clofibrate treatment on cell death was evaluated by the TUNEL assay due to the fact that apoptosis after I/R injury may aggravate myocardial remodeling and decrease cardiac function. Figure 8 shows a statistically significant increase in the number of apoptotic cells in MetS-Sh when compared to Ct-Sh animals. The number of dead cells did not show a significant change among the different groups under I/R conditions; however, a 7-day treatment with clofibrate prevented apoptosis in the MetS group without there being evident change in the Ct group.

## 3. Discussion

MetS is a cluster of risk factors that leads to cardiovascular diseases. MetS doubles the incidence of coronary artery disease; it increases the progression of the atheromatous plaque and is associated with myocardial infarct. The treatment of choice for reducing myocardial ischemic injury is timely and effective myocardial reperfusion; however, reperfusion also adds a further component to myocardial injury [19].

Pro-inflammatory cytokines, growth factors, and several neurohumoral pathways participate in cardiac repair and remodeling after the damage of cardiac tissue evoked by I/R; however, the role of NPs in myocardial damage is still controversial [5]. NPs exert their cardioprotective functions not only as the circulating peptide hormones but also as local autocrine and paracrine factors. The transcriptional regulation of these hormones is less well understood; therefore, the identification of pathways or molecules that regulate NPs’ expression is of potential importance [20,21]. On the other hand, growing evidence suggests that PPARα agonists have a protective function. It has been shown that fibrates administration reverts some of the effects caused by myocardial damage by regulating several processes, such as energy metabolism, oxidative stress, inflammation, and cell differentiation [17,18]. However, as far as we know, the effect of these compounds on NPs’ signaling has not been completely explored. In this work, we show that 7 days of clofibrate pre-treatment is able to diminish the production of pro-inflammatory cytokines, collagen synthesis, and the MMP-2 activation associated with I/R injury. Our study also demonstrates that ANP and ANPr were present in myocardial fibers and mitochondria and were elevated at the same time as inflammation progressed. Here, we also establish a novel role for the PPARα agonist on the regulation of NPs and ANPr expression in hearts from MetS rats subjected to I/R damage.

The results in Table 1 show that clofibrate administration had beneficial effects in the MetS model by reducing body weight and central adiposity. As expected, clofibrate pre-treatment had a triglyceride-lowering effect and increased insulin sensitivity without significantly affecting blood pressure, the concentration of glucose, or total cholesterol. Our data are in line with previous reports in which another PPARα agonist was used [19,22,23].

In this study, we used an I/R model because ischemic damage can be reversed by early reperfusion and because the restoration of blood flow can cause more damage by affecting the myocardial repair and remodeling processes. During myocardial injury due to I/R, inflammation, proliferation, and maturation of cells form part of the reparative response [24]. First, we measured pro-inflammatory cytokine concentrations in the damaged tissue. Figure 1 shows that cytokines were increased in hearts from MetS rats, that there was a larger increase in I/R conditions, and that clofibrate pre-treatment was able to decrease these inflammatory biomarkers. It has been reported that MetS is a pathological condition in which inflammation is present; moreover, TNF-α and IL-1β have been related to the presence of insulin resistance and to the promotion of the pro-inflammatory state by activating NF-κB pathway in a vicious circle [25,26,27]. Regarding the anti-inflammatory effect of clofibrate, our results agree with previous reports by Sun et al. [23] and by our group [18]. PPARα represses inflammation by inhibiting mediators of the NF-κB signaling pathway. It inhibits the production of IL-6 dependent on IL-1 in addition to prostaglandins, such as COX-2, in vascular smooth muscle cells [28]. In addition, it also favors the anti-inflammatory signaling pathway by increasing IκB [29].

Inflammatory cytokines and other growth factors recruit immune cells into the infarcted region in the damaged myocardium, which intensifies the inflammatory reaction and modulates downstream signaling cascades involved in remodeling and reparative responses [30]. Moreover, an imbalance in the equilibrium of synthesis and degradation of extracellular matrix components occurs after I/R injury; hence, we decided to evaluate the levels of fibrosis and the expression of MMP-2 in our model. Our results indicate that the number of infiltrated cells and the accumulation of collagen were increased in MetS and I/R conditions (Figure 2A,B). Clofibrate administration was able to prevent these effects. It is well known that neutrophils and/or macrophages show a strong affiliation to invade the damaged area due to the apoptotic cardiomyocytes and the need for their removal; however, in the present paper, we did not evaluate the cell subpopulations infiltrated [6,30]. Our data are in line with several studies that have demonstrated that MetS signs, such as obesity, insulin resistance, lipotoxicity, and inflammation, play an important role in myocardial fibrosis by activating several downstream signaling cascades [31,32,33]. In addition, scientific evidence has implicated angiotensin II in the inflammatory process as well as in the progression of myocardial fibrosis via binding to AT1 receptor [34]. Regarding this, we previously reported that PPARα agonist (fenofibrate) diminishes the angiotensin II concentrations and AT1 expression in hearts from MetS rats subjected to ischemic injury. Therefore, this might be an additional mechanism to attenuate myocardial damage in our model [18].

Myocardial remodeling is a dynamic process in which MMPs play an important role. MMPs directly activate cytokines, chemokines, cell surface receptors, growth factors, and other proteases, and they contribute to fundamental processes, such as cell proliferation, differentiation, adhesion, migration, and apoptosis. In I/R injury, expression of MMP-2 is of particular interest because MMP-2 degrades extracellular matrix substrates including Type IV collagen, laminin, elastin, interstitial fibrillar collagen, and sarcomeric proteins [35,36]. The expression of this zinc-dependent protease is regulated by mechanical signals, inflammatory factors, hormones, and NPs, among other factors; some authors even suggest that MMP-2 can be used as a possible pharmacological target in the treatment of heart failure [31,36,37,38]. Therefore, we analyzed the expression of MMP-2 in all experimental groups and found that the expression of this enzyme was higher in hearts from MetS rats in sham-operated conditions and under I/R damage without there being changes in the Ct group. Pre-treatment with clofibrate significantly decreased the expression of this protease in both Ct and MetS groups (Figure 3). This protective effect of the PPARα activation by decreasing the expression of MMPs has been previously shown in other experimental models of damage to the myocardium [17,39,40].

On the other hand, there is a strong association between MMP-2 levels with NPs in patients with heart failure [41]. The role of these peptides has been explored in several organs and in pathologies, such as MetS; however, their therapeutic potential and their receptors are just beginning to be expanded [11,42,43]. Hence, we decided to evaluate the effect of clofibrate on the expression of ANP and ANPr in our experimental groups. Results in Figure 4 and Figure 5 demonstrate that the peptide and its receptor were present in higher levels in ventricles from MetS rats subjected to I/R as a compensatory response to injury and that this increase was reduced by pre-treatment with the drug. Our results are in accordance with previous publications that reported that fenofibrate treatment, another PPAR agonist, blocked the increase in plasma levels of NPs in a pig model of heart failure [44]. Some reports have shown that angiotensin II can promote the production of ANP [45,46], and in a previous report, we showed that treatment with fenofibrate decreases the levels of this peptide [18]; therefore, this might be another synergic way by which clofibrate could decrease the ANP levels in our experimental model.

Relatively little information exists regarding the regulation of the ANPr expression. Our findings demonstrate a positive association between the ANP and ANPr in ventricles from MetS rats subjected to I/R injury, which supports the results obtained by Pandey et al. [47] showing that NPs may regulate ANPr gene expression. Our work suggests that the activation of PPARα was able to regulate the expression of this cardiac receptor; however, further studies are needed to evaluate the binding of this transcriptional factor to the receptor gene. Moreover, several studies have shown that NPs may exhibit protective effects in I/R injury through mitochondria-mediated mechanisms.

Some authors have reported that NPs provide myocardial protection by regulating mitochondrial biogenesis and swelling, the production of reactive oxygen species, the oxidation of fat, and the synthesis of ATP, leading to decreased cell death [48,49,50]. The representative electron micrographs presented in Figure 7 demonstrate that hearts damaged by I/R from MetS animals showed significantly higher levels of ANP and ANPr in myocardial fibers and mitochondria compared to the Ct group. This effect was abolished by treatment with clofibrate. However, additional studies are required to elucidate the pathophysiological significance of the presence of these factors in mitochondrial dynamics and function during myocardial damage in MetS rats.

Currently, BNP and NT-proBNP are widely used as diagnostic biomarkers for heart failure and myocardial infarction in clinical medicine [51]. Cardiac myocytes constitute a major source of BNP-related peptides in response to cardiac wall stress of the left ventricle in acute myocardial infarction. BNP is synthesized first as a 108 amino acid prohormone (proBNP), and proBNP is then cleaved by furin into the biologically active 32 amino acid BNP (C-terminal fragment) and into the biologically inactive 76 amino acid N-terminal fragment (NT-proBNP). In our model, we decided to measure NT-proBNP rather than BNP because of its higher half-life (120 min vs. 20 min) [51,52]. We observed no statistically significant difference of NT-proBNP among groups (I/R or MetS conditions compared to Ct-Sh) (Figure 6). We hypothesized that we would have possibly observed differences in BNP among the different groups because BNP is biologically active and its biological actions counteract the fibrosis and inflammation associated with myocardial infarction in rats [52,53,54]. Additionally, BNP and NT-proBNP diagnostic utility in clinical studies has been researched by measuring both biomarkers in circulation (serum or plasma) [51]; thus, we can possibly observe differences in BNP peptides among groups when measuring circulating NT-proBNP and/or BNP. Therefore, we need to compare in situ and circulating levels of BNP and NT-proBNP concentrations among our different groups to test this hypothesis. On the other hand, a small quantity of BNP is stored in cardiomyocyte granules; instead, BNP is transcribed as needed in response to wall stress and once proBNP is released into circulation, it is cleaved to NT-proBNP and finally BNP [55,56]. Therefore, the lack of statistically significant differences in NT-proBNP in our groups might be also related to the small amounts of BNP stored in cardiomyocytes; however, additional studies are needed to test this hypothesis.

Since ANP and BNP have similar affinities for ANPr [46] and in our model we observed that ANPr was upregulated in I/R groups, we hypothesized that BNP might act locally by ANPr binding to reduce myocardial fibrosis. We need to carry out ligand binding assays to verify this hypothesis; however, this is outside the objectives of this study. Factors including endocrine and paracrine modulation by other neuro-hormones and cytokines are also of importance. Our findings support the idea to perform assays of BNP and other biomarkers, such as creatine kinase, cardiac troponin I, heart-type fatty acid-binding protein, adrenomedullin, and osteoprotegerin, in combination with imaging analysis as a tool for differential diagnosis and therapy in heart diseases [57,58].

Finally, we evaluated the clofibrate effect on cell death due to the fact that an excessive inflammatory response after I/R injury induces myocardial apoptosis and that apoptosis may aggravate the formation of cardiac scars, leading to decreased cardiac function [59]. Figure 8 shows that MetS is associated with an increase in cardiac apoptosis when compared to the Ct group. This increase was higher when hearts were subjected to I/R conditions. The 7-day pre-treatment with clofibrate reverted this effect. Our data are in accordance with previous studies that showed the association of apoptosis and heart failure and the anti-apoptotic role of PPARα agonists [16,17,39].

## 4. Materials and Methods

### 4.1. Experimental Animals

All animal protocols and procedures were performed in compliance with the recommendations of the official normative for laboratory animal care protocols (SAGARPA, NOM-062-ZOO-1999, Mexico). Male Wistar rats, 25 days old, were randomly separated into two groups of 12 animals: group 1, control (Ct) rats that were given tap water for drinking, and group 2, MetS rats that received 30% sucrose in their drinking water for 24 weeks. The animals were kept under 12 h light/obscurity cycles and environmental temperature ranging from 18 to 26 °C. They were fed commercial rodent pellets (PMI Nutrition International Inc., LabDiet 5008, Richmond, IN, USA) ad libitum [18].

Next, both Ct and MetS animals were randomly subdivided into two equal groups according to receive vehicle (Vh) or clofibrate (Clo, 100 mg/kg/day) by intraperitoneal injection every day for 7 days [16,17]. At the end of the treatment, the animals were weighed, and systolic arterial blood pressure was determined in conscious animals by the tail-cuff plethysmography technique previously performed [18]. The intra-abdominal white adipose tissue (retroperitoneal fat pad) was also carefully dissected with scissors after euthanasia, wet weight was determined, and then the tissue was discarded.

### 4.2. Ischemia Reperfusion Model

Animals were anesthetized for ischemia/reperfusion (I/R) surgery with an intramuscular drug mixture of ketamine hydrochloride 80mg/kg (from Laboratorios Aranda, Queretaro, Qro. Mexico) and xylazine hydrochloride 10mg/kg (from PiSA Farmaceutica, Guadalajara, Jal., Mexico). The I/R procedure was performed as previously reported by Oidor Chan et al. [22] with a modification in the time of reperfusion. After inducing sedation and maintaining mechanical ventilation (70 breaths per minute; vol. 8–10 mL/kg), asepsis of the thoracic area was performed. The chest was opened by a lateral incision along the upper margin of the fourth rib to expose the heart. A PE-10 tube was inserted between the suture and anterior descending coronary artery (LAD) immediately before ligation. After 30 min of coronary artery ligation, we removed the PE-10 tubing to establish reperfusion for 60 min. In sham (Sh), control, and MetS groups, the procedure was identical, except the left anterior descending artery was not transiently ligated.

### 4.3. Serum Biochemical Variables

We determined the basal serum biochemical parameters in animals that were fasted overnight. The blood samples were collected by vessel puncture from the vena cava. Serum was isolated by centrifugation and stored until needed. The baseline fasting values of glucose, total cholesterol, HDL cholesterol, LDL cholesterol, and triglycerides were measured with commercial enzymatic kits (RANDOX Laboratories Ltd., Crumlin, Country Antrim, UK); insulin was determined by a rat-specific insulin radioimmunoassay (Linco Research, Inc., Saint Charles, MO, USA), as previously reported [60].

### 4.4. Cardiac Cytokines and NT-proBNP Quantification

At the end of the surgery procedure, left ventricle samples were obtained and homogenized with lysis buffer (50 mM HEPES, pH 7.5; 150 mM NaCl, 1% glycerol, 1% Triton X-100, 1.5 mM MgCl_2_ and 5mM EGTA, 1mM PMSF) and protease inhibitors cocktail. Later, the homogenates were centrifuged at 10,000× *g* and the protein quantification was performed with bicinchoninic acid technique (Pierce BCA protein assay kit, Thermo Scientific, Rockford, IL, USA). The cytokine values were obtained by the ELISA sandwich technique as previously reported [17].

N-terminal pro-brain natriuretic peptide (NT-proBNP) concentrations were assayed in left ventricle homogenates using a Rat NT-proBNP ELISA kit (MyBioSource, San Diego, CA, USA) according to the manufacturer’s instructions.

### 4.5. Western Blot

Frozen left ventricle from the different experimental groups was homogenized with lysis buffer pH 7.4 (250 mM Tris-HCl, 2.5 mM EDTA) and protease inhibitors cocktail (Complete^®^ tablets, Roche Applied Science, Mannheim, Germany) using a tissue homogenizer (Fisher Scientific, Waltham, MA, USA) at 4 °C. The homogenate was centrifuged at 5000× *g* for 10 min at 4 °C and the supernatant was separated and stored at −70 °C until required. Total protein was determined by the Bradford method [61]. One hundred µg of protein from all experimental groups was separated in SDS-PAGE gel (10%) and electro transferred to polyvinylidene difluoride (PVDF) membrane. The blots were blocked with 5% non-fat dehydrated milk for three hours under continuous stirring at room temperature. Later, the membranes were incubated overnight with primary antibodies to MMP-2 from Santa Cruz Biotechnology (Santa Cruz, CA, USA) at 4 °C and subsequently with its corresponding secondary antibody for three hours at room temperature (Jackson ImmunoResearch, Suffolk, UK). All the blots were incubated with β-actin as a loading control. After incubation, plates were revealed with a peroxidase-chemiluminescent kit (Clarity western ECL substrate, Bio-Rad Laboratories, Inc. Hercules, CA, USA). The bands were quantified by densitometry employing the Quantity One software 4.6 (Bio-Rad Laboratories, Inc. Hercules, CA, USA) by using a GS-800 densitometer. The results are expressed as arbitrary units (AU) of the ratio between MMP-2 and β-actin [60].

### 4.6. Histology

A fragment of the left ventricle wall obtained from 3 rats from each group, previously preserved at −70 °C, was gradually thawed (consecutive changes to −20 °C, 4 °C, and room temperature). Then, the tissue was fixed in 4% paraformaldehyde (PFA) for 48 h, with 5 washes (15 min each) with PBS [62]. Subsequently, cryoprotection was performed by placing the tissues from each group in 30% sucrose for 48 h [63,64]. Frozen cross-sections at 10 μm were mounted on gelatinized or electrocharged slides. The slides were kept at 4 °C until required.

### 4.7. Hematoxylin–Eosin Stain (HE) and Picrosirius Red (PSR)

HE and PSR staining were performed in the gelatinized slide sections. Slides containing cardiac tissue sections (2 blocks with 3 heart fragments, from 3 different rats from each group on each slide) were selected, dried at room temperature for one hour, and rehydrated for 30 min before use. HE staining (ab245880, Abcam PLC, Cambridge, UK) was performed to visualize the morphology of the tissue as well as the cellular infiltrates. First, most of the water from the cuts was removed and 300 μL of hematoxylin, Mayer’s (Lillie’s Modification) was applied to completely cover each tissue section and incubated for 5 min. The slides were rinsed in two changes of distilled water to remove excess stain; we applied adequate Bluing Reagent to completely cover each tissue section and incubated for 10–15 s; we rinsed the slides in two changes of distilled water and we immersed the slides in absolute alcohol to blot excess off. Then, we applied 300 μL Eosin Y Solution (Modified Alcoholic) to completely cover each tissue section to excess and incubated for 2–3 min; the slides were rinsed using absolute alcohol and dehydrated in three changes of absolute alcohol. Finally, the slides were mounted in synthetic resin. Myocardial collagen volume fraction (CVF) was measured by picrosirius red (PSR) staining (ab150681, Abcam PLC, Cambridge, UK) [65,66]. Slides containing cardiac tissue sections (2 blocks with 3 heart fragments, from 3 different rats from each group on each slide) were selected, dried at room temperature for one hour, and rehydrated for 30 min before use. The water from the slides was removed and 350 μL of PSR Solution was applied to completely cover the tissue section and incubated for 60 min; the slides were rinsed quickly in two changes of acetic acid solution. We rinsed the slides in absolute alcohol and dehydrated in two changes of absolute alcohol. Finally, the slides were mounted in synthetic resin. Visualization of HE and PSR staining were performed by light microscopy (Olympus BX51) and PSR was detected on a Floid Cell Imaging Station (Life Technologies, Carlsbad, CA, USA). The number of infiltrated cells in HE images was quantified and FVC values were calculated (ventricular collagen area/field area) [65] from the measurement of the corresponding areas in PSR images acquired at 20× employing Image-Pro Premier 9.0 (Media Cybernetics) software.

### 4.8. Immunodetection of Atrial Natriuretic Peptide (ANP) and Atrial Natriuretic Peptide Receptor (ANPr)

Electrocharged slide sections containing cardiac tissue sections were chosen, dried at room temperature for one hour, passed through xylol–alcohol until rehydrated in PBS, and re-fixed with 4% PFA (10 min). The fluorescence generated by aldehydes was quenched by incubation in a solution of glycine 0.1M/PBS pH 7.4 for 2 min. Antigen retrieval was performed with Tris (0.5M)/EDTA (0.1M) solution, pH 9 at 95 °C for 10 min. The sections were washed with PBS 5 times, depositing and removing sufficient volume on the sections with a micropipette. They were then permeabilized with TritonX-100 0.2%/BSA 1.5% in PBS. Each tissue block was surrounded with a hydrophobic pencil. After incubation with the blocking solution (BS) (BSA 3%/ 0.1 triton X-100, in PBS) at room temperature for 30 min in a humid chamber [67], the sections were incubated with the corresponding primary antibody for ANP (4.77 μg/mL (ab225844, Abcam PLC, Cambridge, UK)) or ANPr (2.5 μg/mL (ab14356, Abcam PLC, Cambridge, UK)) for 72 h at 4 °C in the dark and in a humid chamber [64]. Once this period was over, the sections were washed with PBS (4× 10 min), depositing and removing sufficient volume on the sections with a micropipette. Incubation with secondary antibody (10μg/mL (ab150079, Abcam PLC, Cambridge, UK)) in BS for 1.5 h at room temperature was performed. Staining with 4’,6-diamidino-2-Phenylindole (DAPI) (1.43 μM) was used to visualize the nuclei, and they were mounted with mounting medium for fluorescence. Nonspecific binding was verified by using negative controls, which were incubated with the blocking solution (3% BSA/0.1 triton X-100, in PBS) without the primary antibody. Immunofluorescence assays were performed using previously reported methods with small modifications [68,69].

### 4.9. TUNEL Test

Once the PNA immunoassay was completed, the reaction mixture containing the enzyme solution (TdT) and label solution (fluorescein-dUTP) corresponding to the TUNEL assay (In Situ Cell Death Detection Kit, Fluorescein (Roche Applied Science, Mannheim, Germany, 11684795910)) was applied [64,70,71]. The slides were incubated at 37 °C for 60 min in the dark inside a humid chamber. At the end of incubation, the sections were washed with PBS and the sections were mounted with mounting medium and analyzed by fluorescence microscopy. Tissues fixed, permeabilized, and treated with label solution (without terminal transferase) instead of TUNEL reaction mixture were used as negative controls; as positive controls, we used tissues incubated with DNase I recombinant (3000 U/mL-10 min-room temperature). TUNEL-positive cells were quantified and the percentage of positive cells (marked with DAPI) per total cell number per field was calculated.

### 4.10. Image Acquisition, Number of Cell Infiltrate, Positive TUNEL Cells, and ANP and ANPr Intensity Quantification

Image acquisition for light microscopy para HE was performed with a Q-Imaging camera (Microplublisher 5.0 Real-Time Viewing (RTV) coupled to an Olympus BX51 microscope. Visualization and acquisition of fluorescent images for PSR, ANP, ANPr, and TUNEL were performed using Floid Cell Imaging Station equipment. The number of infiltrated cells (HE staining), positive TUNEL cells with respect to the total number of cells (DAPI) per field, and the quantification of fluorescence intensity per area unity (integrated optical density (IOD) (lum/pix^2)) to ANP and ANPr were carried out with the Image-Pro Premier 9 (Media Cybernetics). At least 4 fields (20X) of each animal (3 per group) were quantified, with a total of at least 12–24 determinations per condition.

### 4.11. Immune Colloidal Gold Technique

The samples were processed as reported by Soria Castro et al. [72] with small modifications. The hearts were excised and divided for histological analyses while fresh; small pieces of rat heart left ventricle were fixed for 2 h in 4% paraformaldehyde (cat 30525-894. Electron Microscope Science, Haffield, PA, USA) and 0.1% glutaraldehyde (cat.111-30-8; Electron Microscope Science) in 0.1M PBS, pH 7.4. The samples were dehydrated in gradual alcohols and embedded in LR White resin (cat.14381; Electron Microscope Science, Haffield, PA, USA). Ultrathin sections (50 nm) were placed on carbon/formvar-coated nickel grids of 150 mesh. The grids were floated on droplets of PBS for 10 min and then whole goat serum for 1 h. Next, the grids were washed with droplets of PBS 10 min 3 times, after which each grid was floated on 30 μL of antibodies against the ANP (cat. ab225844 Abcam Cambridge, UK) and the ANP receptor ((ab14356), Abcam, Cambridge, UK) at a dilution of 1:20 overnight at 4 °C in a wet chamber. The grids were then washed with droplets of PBS for 10 min 3 times. Finally, the expressions were displayed with anti-rabbit IgG antibody conjugated to 15 nm gold particles (25112 Electron microscopy sciences, Hatfield, PA, USA). The grids were first stained with 2% uranyl acetate and then lead citrate, washed with water, and air dried. The sections were observed in an electron microscope (Model JEM-1011) at 80 kV (JEOL Ltd., Tokyo, Japan) and analyzed with AMT-5.42.391 software. Ten random fields were taken over a total area of 62.5 microns at ×50,000 magnification.

### 4.12. Statistical Analysis

The program GraphPad Prism version 9.4 (GraphPad software, La Jolla, CA, USA) was used to generate graphs and to perform statistical analyses. Results are expressed as mean ± standard error of the mean (SEM). Statistical significance was determined by one-way ANOVA followed by Tukey’s post-hoc test and Kruskal–Wallis if the data were normally distributed or not. The differences were considered when *p* < 0.05.

## 5. Conclusions

The PPARα agonist clofibrate has a beneficial cardiac effect on ischemia/reperfusion conditions by decreasing the inflammatory response that, in turn, decreases myocardial fibrosis, the MMP-2 expression, and apoptosis. As a consequence of these processes, the ANP/ANPr signaling is diminished as a compensatory response. Our study provides important insight into the regulation of both ANP and ANPr in the hearts of MetS rats by clofibrate.

## Figures and Tables

**Figure 1 ijms-24-05321-f001:**
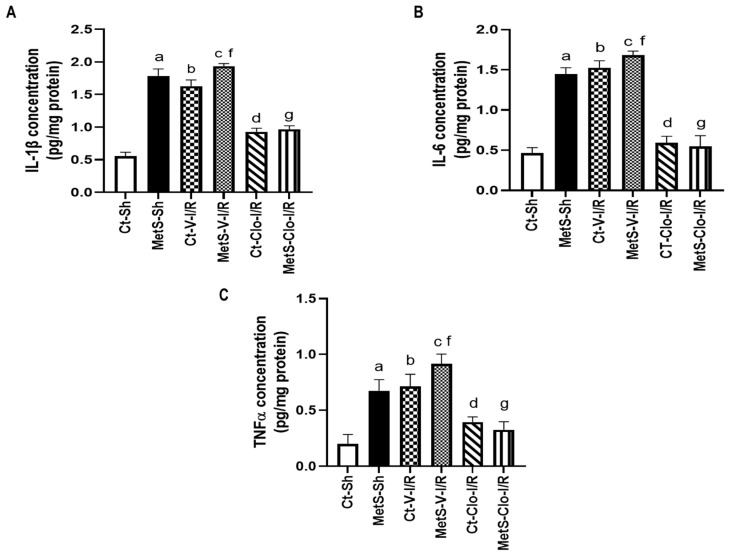
Effect of PPARα agonist clofibrate in the production of pro-inflammatory interleukins that are induced by ischemia/reperfusion injury in hearts from metabolic syndrome rats. (**A**) IL-1β, (**B**) IL-6, and (**C**) TNF-α concentration in left ventricles from control and MetS rats. Data represent mean ± SEM (n = 6 per group). ^a^ *p* < 0.0001 vs. Ct-Sh; ^b^ *p* < 0.0001 vs. Ct-Sh; ^c^ *p*< 0.0001 vs. Ct-V-I/R; ^d^ *p* < 0.0001 vs. Ct-V-I/R; ^f^ *p* < 0.0001 vs. MetS-Sh; ^g^ *p* < 0.0001 vs. MetS-V-I/R. Abbreviations: Ct = control; MetS = metabolic syndrome; Sh = sham-operated rats; V = vehicle; Clo = clofibrate.

**Figure 2 ijms-24-05321-f002:**
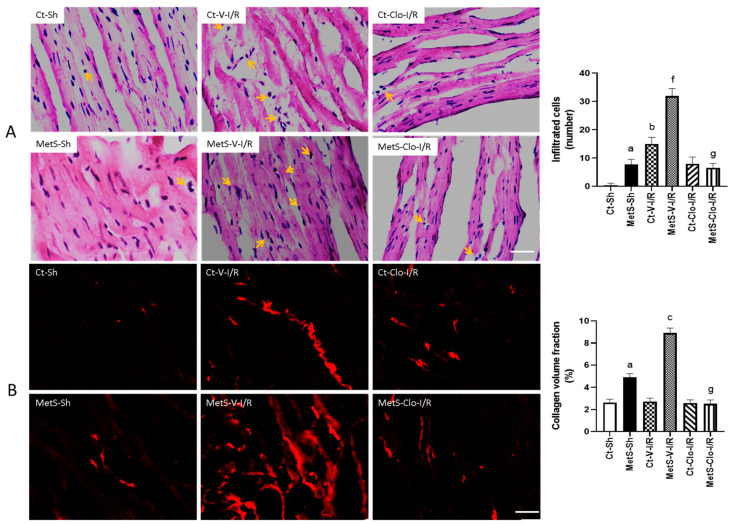
Effect of treatment with clofibrate on infiltrated cells and collagen volume fraction (CVF) in hearts with ischemic/reperfusion (I/R) damage. In (**A**), the sites where cellular infiltrates are located are indicated with arrows in images stained with hematoxylin–eosin (HE). In (**B**), the fluorescence areas with collagen deposits are distinguished in red with picrosirius red (PSR) staining. Representative images of histological examinations are presented and the mean ± SEM values of the number of infiltrated cells and the mean ± SEM of CVF % are shown in the graphs. ^a^ *p* < 0.0001 vs. Ct-Sh; ^b^ *p* < 0.0001 vs. Ct-Sh; ^c^ *p* < 0.0001 vs. Ct-V-I/R; ^f^ *p* < 0.0001 vs. MetS-Sh; ^g^ *p* < 0.0001 vs. MetS-V-I/R. Abbreviations: Ct = control; MetS = metabolic syndrome; Sh = sham-operated rats; V = vehicle; Clo = clofibrate. Bar = 100 μm.

**Figure 3 ijms-24-05321-f003:**
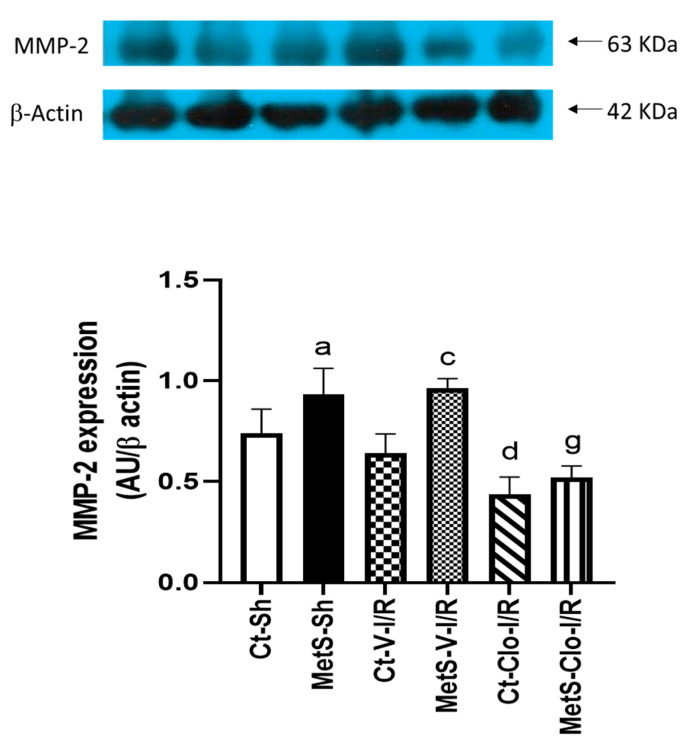
Effect of clofibrate treatment in the expression of MMP-2 in damaged ventricles from control and MetS rats. Expression was evaluated by Western blot in the myocardial ischemic area from sham (Sh), I/R-V, and I/R-Clo groups. Data represent mean ± SEM normalized to β-actin (n = 6 rats per group). ^a^ *p* < 0.0001 vs. Ct-Sh; ^c^ *p* < 0.0001 vs. Ct-V-I/R; ^d^ *p* < 0.0001 vs. Ct-V-I/R; ^g^ *p* < 0.0001 vs. MetS-V-I/R. Abbreviations: Ct = control; MetS = metabolic syndrome; Sh = sham-operated rats; V = vehicle; Clo = clofibrate.

**Figure 4 ijms-24-05321-f004:**
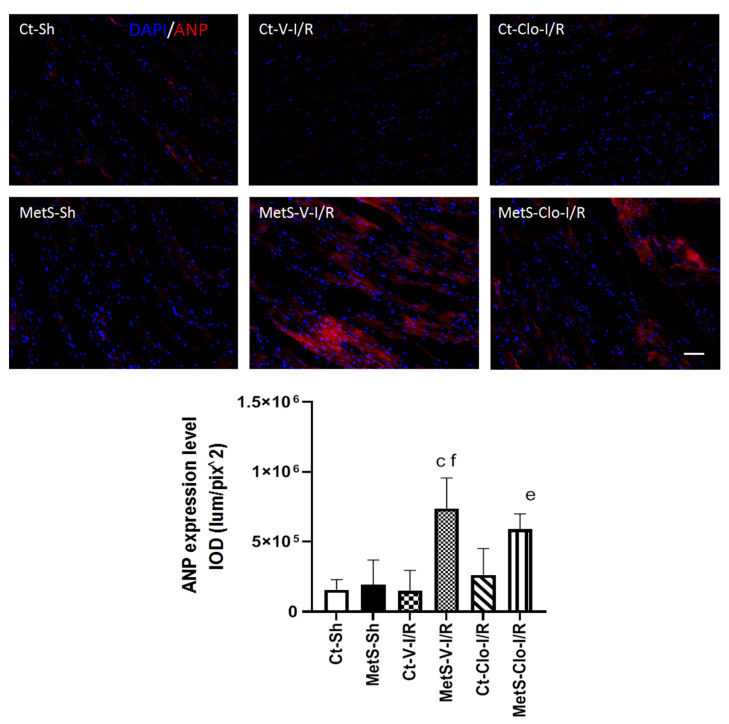
Effect of clofibrate administration on the expression of atrial natriuretic peptide (ANP) in the left ventricles from control and MetS rats. The representative images show the immunodetection of ANP (red) and 2-[4-(Aminoiminomethyl) phenyl]-1H-Indole-6-carboximidamide hydrochloride (DAPI) was used to label the nuclei. The graph showing the expression of immunodetection levels is presented. Data represent mean ± SEM. ^c^ *p* < 0.0001 vs. Ct-V-I/R; ^e^ *p* < 0.0001 vs. Ct-Clo-I/R; ^f^ *p* < 0.0001 vs. MetS-Sh. At least 4 fields of each animal (3 rats per group) were quantified, with a total of at least 12–24 determinations. Abbreviations: Ct = control; MetS = metabolic syndrome; Sh = sham-operated rats; V = vehicle; Clo = clofibrate. Bar = 100 μm. IOD = integrated optical density, corresponding to ANP expression level.

**Figure 5 ijms-24-05321-f005:**
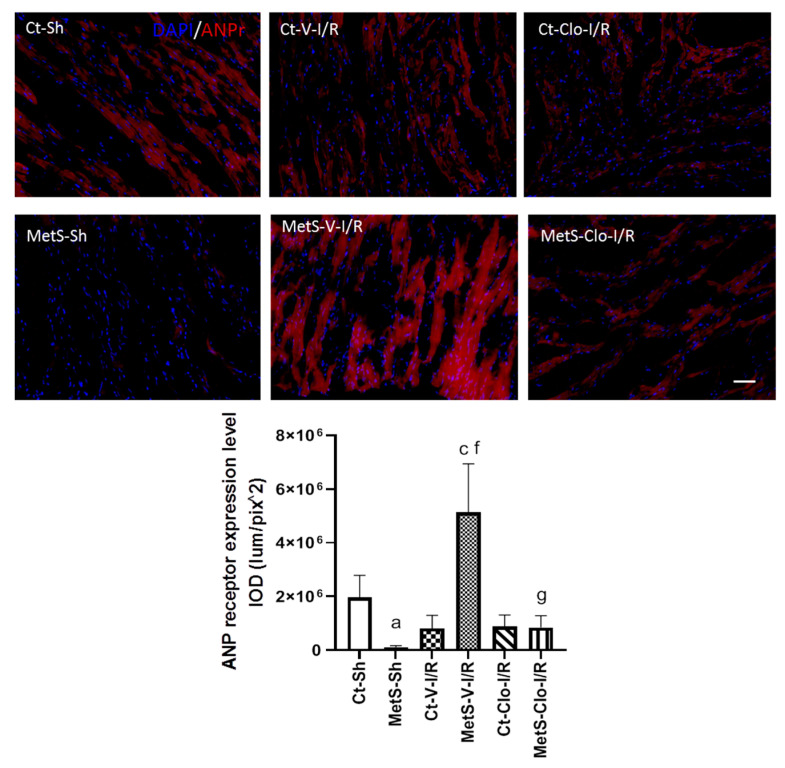
Effect of clofibrate treatment in the atrial natriuretic peptide receptor (ANPr) expression in left ventricles from MetS rats under ischemia/reperfusion conditions. Immunodetection of ANPr is shown in red and the nuclei are marked with DAPI. The graph shows the receptor expression levels. Data represent mean ± SEM. ^a^ *p* < 0.0001 vs. Ct-Sh; ^c^ *p* < 0.0001 vs. Ct-V-I/R; ^f^ *p* < 0.0001 vs. MetS-Sh; ^g^ *p* < 0.0001 vs. MetS-V-I/R. At least 4 fields of each animal (3 rats per group) were quantified, with a total of at least 12–24 determinations. Abbreviations: Ct = control; MetS = metabolic syndrome; Sh = sham-operated rats; V = vehicle; Clo = clofibrate; Bar = 100 μm; IOD = integrated optical density, corresponding to ANPr expression level.

**Figure 6 ijms-24-05321-f006:**
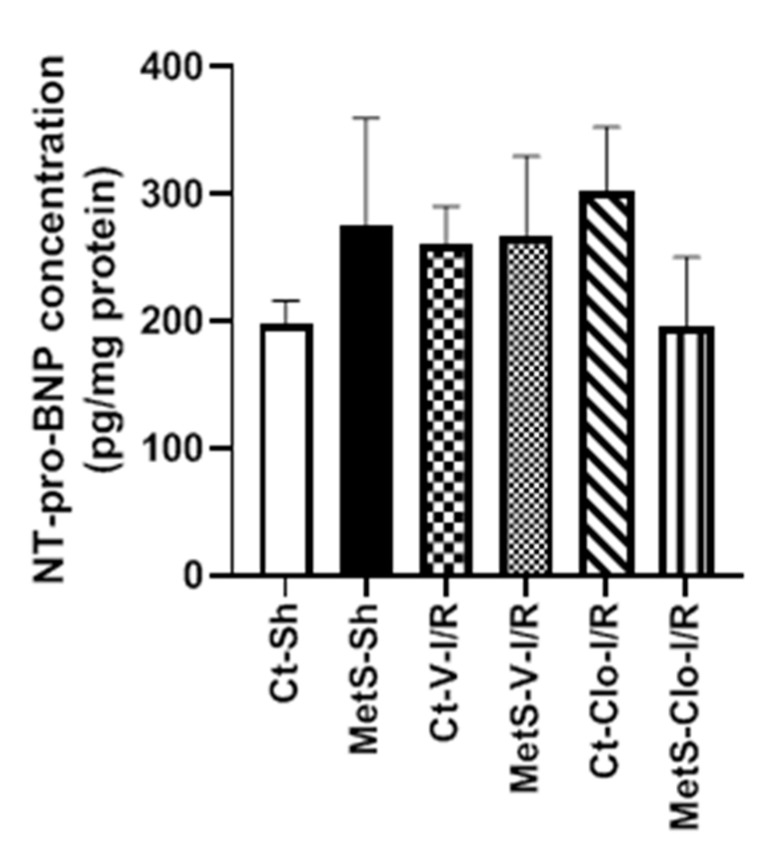
Tissue levels of N-terminal pro-B-type natriuretic peptide (NT-proBNP) in control and metabolic syndrome rats subjected to ischemia/reperfusion damage and pre-treated with clofibrate. Values are mean ± SEM. N = 5 per group; Abbreviations: Ct = control; MetS = metabolic syndrome; Sh = sham-operated rats; I/R: ischemic reperfusion; V = vehicle; Clo = clofibrate.

**Figure 7 ijms-24-05321-f007:**
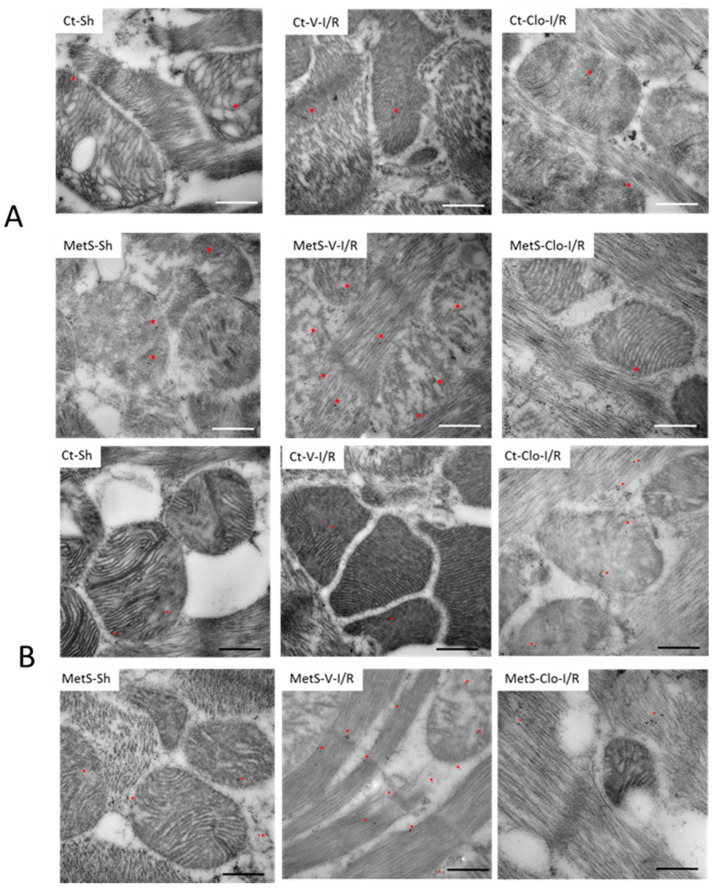
Expression of atrial natriuretic peptide (ANP) and atrial natriuretic peptide receptor (ANPr) in fibers and mitochondria of hearts from MetS rats under ischemia/reperfusion injury. Representative immune-electron micrograph for ANP (**A**) and ANPr (**B**) of left ventricles from control (Ct) and metabolic syndrome (MetS) rats subjected to sham (Sh) or ischemia/reperfusion (I/R) damage and treated with either vehicle (V) or clofibrate (Clo). The images show the signal (red asterisk) obtained by 15 nm gold particles in the different treatments. Ultrastructural alterations in fibers and mitochondrial architecture in I/R-damaged groups are also evident, such as amorphous matrix densities and severe mitochondrial swelling. Bar = 500 nm.

**Figure 8 ijms-24-05321-f008:**
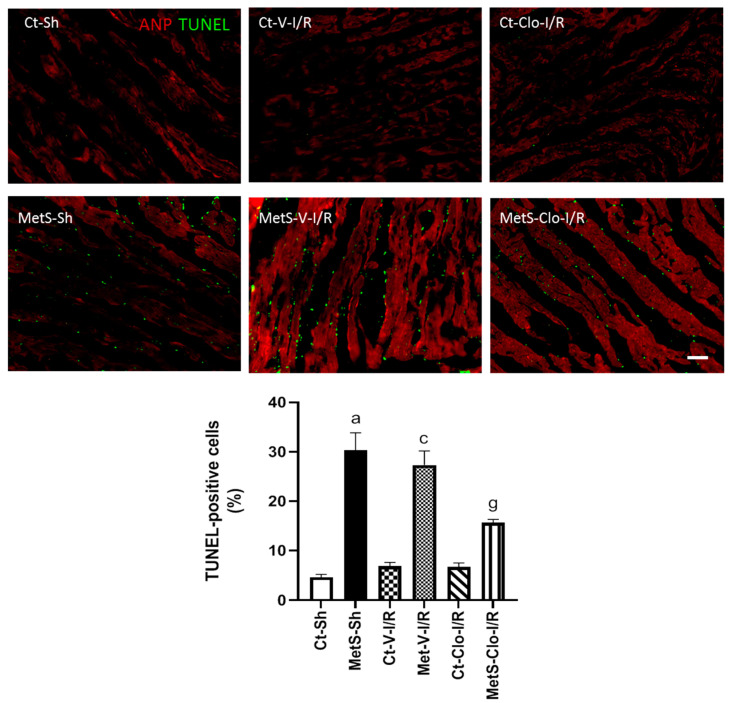
PPARα agonist treatment decreased apoptosis in ischemic/reperfused (I/R) hearts from MetS rats. The TUNEL assay was performed on cardiac tissue sections previously subjected to ANP immunodetection in red; this facilitates the visualization of positive TUNEL nuclei. The graph with the percentage of TUNEL-positive cells with respect to the total of DAPI-labeled nuclei in each group is presented. ^a^ *p* < 0.0001 vs. Ct-Sh; ^c^ *p* < 0.0001 vs. Ct-V-I/R; ^g^ *p* < 0.0001 vs. MetS-V-I/R. At least 4 fields of each animal (3 rats per group) were quantified, with a total of at least 12–24 determinations. Abbreviations: ANP = atrial natriuretic peptide; Ct = control; MetS = metabolic syndrome; Sh = sham-operated rats; V = vehicle; Clo = clofibrate. Bar = 100 μm.

**Table 1 ijms-24-05321-t001:** Effect of clofibrate on the general characteristics and serum parameters of control and MetS rats.

	Ct-V	Ct-Clo	MetS-V	MetS-Clo
Body weight (g)	536.5 ± 16.0	486.8 ± 16.9	559.6 ± 19.4	440.5 ± 16.9 ^b^
Systolic blood pressure (mm Hg)	98.9 ± 3.6	97.8 ± 2.4	143.6 ± 4.9 ^a^	139.7 ± 9.3 ^a^
Intra-abdominal fat (g)	5.33 ± 1.3	3.6 ± 0.4	16.6 ± 3.4 ^a^	7.7 ± 0.9 ^b^
Triglycerides (mg/dL)	71.2 ± 16.0	59.1 ± 7.8	150.3 ± 12.2 ^a^	84.5 ± 6.1 ^b^
HDL-c (mg/dL)	40.1 ± 5.3	38.6 ± 4.9	25.8 ± 2.8 ^a^	27.6 ± 3.2
Non-HDL-c (mg/dL)	19.8 ± 1.7	10.5 ± 1.2 ^b^	36.7 ± 3.6 ^a^	11.9 ± 3.2 ^b^
Total cholesterol (mg/dL)	53.0 ± 5.7	54.2 ± 5.2	51.7 ± 5.4	49.3 ± 6.9
Glucose (mM)	5.1 ± 0.22	4.9 ± 0.14	5.8 ± 0.08	5.3 ± 0.4
Insulin (ng/mL)	0.063 ± 0.003	0.076 ± 0.001	0.36 ± 0.09 ^a^	0.12 ± 0.02 ^b^
HOMA-IR	0.87 ± 0.2	0.67 ± 0.09	3.7 ± 0.5 ^a^	1.0 ± 0.8 ^b^

Values are mean ± SEM. Abbreviations: Ct = control; MetS = metabolic syndrome; V = vehicle; Clo = clofibrate; HOMA-IR = homeostatic model assessment of insulin resistance; HDL-c: high-density lipoprotein cholesterol; n = 6 animals per group; ^a^ *p* < 0.05 MetS vs. Ct same treatment; ^b^ *p* < 0.05 against vehicle corresponding group.

## Data Availability

The data in our study are available from the corresponding author upon reasonable request.

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
