# Peer review of "PPAR Alpha Activation by Clofibrate Alleviates Ischemia/Reperfusion Injury in Metabolic Syndrome Rats by Decreasing Cardiac Inflammation and Remodeling and by Regulating the Atrial Natriuretic Peptide Compensatory Response"

_ijms, 2023, doi:10.3390/ijms24065321_

Round 1

Reviewer 1 Report (Previous Reviewer 1)

In previously submitted manuscript: ijms-2008234,  I have reviewed original/revised  version. In this manuscript , please provide clear uncropped/Unedited blots for WB.

Author Response

R= Thank you for your observation. In the revised version of our manuscript, we provided the original western blot images.

Reviewer 2 Report (Previous Reviewer 2)

The manuscript, as it stands, is unpublishable. This is because the authors are unwilling or unable to improve the writing and presentation of the data.

Author Response

We regret the harsh comments from reviewer 2. In a previous version of our manuscript (ijms-2008234), we have followed most of his/her recommendations in order to improve the manuscript, but the organization and presentation of results are difficult to change as a whole.

Reviewer 3 Report (New Reviewer)

The present article presents interesting data on the effect of clofibrate on cardiac status in I/R rats with metabolic syndrome. The article is well written, but there are some questions and comments.

Line 56-61. Second sentence is an alien here.

Line 101 – receptor, receptors

Why clofibrate was chosen for this research?

Line 118 – Ct rats - the transcription follows only further down in the table, you need to specify here

Table 1. Please indicate what type of blood pressure. Wistar rats typically has a systolic blood pressure in the range of 110-120 mmHg, why in your case it is so low in control groups?

In rodents, HDL is thought to be the main cholesterol carrier and there should be a correlation between lipoprotein and total cholesterol concentrations in the blood. In your case, there is a decrease in lipoproteins in the MetS groups, whereas total cholesterol levels have the same value everywhere. How can this be explained?

In the results section it would be good to make a transition to the results of the experiment on I/R, in the present form it is not immediately clear what we are talking about.

Fig. 1, 3, 5, 7 – the caption looks like as a conclusion but not a caption.

Line 139 – HE and PSK – no transcription given

Line 147 – maybe disactivated?

Discussion. The first sentence is a repetition of the following text, so it can be deleted.

Line 267. Authors says “lipidemic-lowering effect”, in the results it is clearly demonstrated only TG lowering, so maybe it will be better to change the phrase into TG lowering?

Line 295. It seems “It” is missing here. Also, the reference is needed in this sentence.

Author Response

The present article presents interesting data on the effect of clofibrate on cardiac status in I/R rats with metabolic syndrome. The article is well written, but there are some questions and comments.

Line 56-61. Second sentence is an alien here.

R= Thank you for your observation. We deleted the sentence.

Line 101 – receptor, receptors

R= Thank you for your observation. We corrected the sentence.

Why clofibrate was chosen for this research?

R= We consider several properties of the drug such as: its role as a specific PPAR alpha agonist, the effects of clofibrate at the cardiac and vascular level in different models; and finally, our group previously reported that clofibrate treatment, given as a pre-treatment or immediately after myocardial infarction decreases myocardial ischemia-induced inflammation. We are citing the corresponding reference [17].

Line 118 – Ct rats - the transcription follows only further down in the table, you need to specify here

R= Thank you for your observation. We added the definition.

Table 1. Please indicate what type of blood pressure. Wistar rats typically has a systolic blood pressure in the range of 110-120 mmHg, why in your case it is so low in control groups?

R= Thank you for your observation. We determined the systolic blood pressure and we have included this information in the methods section and in the table 1. The values of blood pressure were reported as the mean from 6 animals; the final recorded blood pressure for a given rat was the average of four to six successive readings. The blood pressure values reported in the present study were consistent with those described in previous reports by our group.

In rodents, HDL is thought to be the main cholesterol carrier and there should be a correlation between lipoprotein and total cholesterol concentrations in the blood. In your case, there is a decrease in lipoproteins in the MetS groups, whereas total cholesterol levels have the same value everywhere. How can this be explained?

R= The reviewer was right in her/his observation. There are fundamental differences in the plasma lipoprotein profile of the rodents relative to the human profile. Whereas humans carry 75% of plasma cholesterol in LDL, mice carry 80% of plasma cholesterol in HDL. This major difference can be attributed to the lack of expression of cholesterol ester transfer protein and to the higher amount of apo B48-containing lipoproteins. Further, HDL particles contain substantial amounts of ApoE. However, the mechanism involved that promotes uptake of both free and esterified cholesterol from HDL, seems to be qualitatively similar in mouse to human, this is through the SR-.BI. (DOI: 10.1186/1743-7075-3-17) (doi: 10.1016/j.bbalip.2016.03.00) (10.2174/1389201011208062435). (10.2174/1389201011208062435)

Metabolic syndrome is characterized by a pathological state of metabolic disorders. In MetS a state of hyperinsulinemia, hypertriglyceridemia associated with a decrease in HDL-C has been described. Under these conditions, the VLDL are responsible for transporting the largest amount of lipids such as cholesterol and triglycerides. An explication is the massive arrival of non-esterified fatty acids (NEFA) to the hepatocyte becomes a direct stimulus for the production of VLDL, together with an increased synthesis of cholesterol. In this context, HDL also present modification in their composition by increasing the content of triglycerides and reducing apo A-I levels. As consequence, the catabolism of HDL catabolism increases and hence their blood levels decrease.

Moreover, it has been reported that total cholesterol to HDL-C ratio is a strong predictor for the incidence of cardiovascular disease (DOI: 10.13057/nusbiosci/n010201; doi:10.1088/1742-6596/1524/1/012126). Regarding this information, our results showed that the MetS rats present a higher TC:HDL-C ratio when compared to Control rats, so the risk of cardiovascular disease was higher.

In the results section it would be good to make a transition to the results of the experiment on I/R, in the present form it is not immediately clear what we are talking about.

R= Thank you for your suggestion. We have reworked some sections from results according to reviewer suggestion.

Fig. 1, 3, 5, 7 – the caption looks like as a conclusion but not a caption.

R= Thank you for your suggestion. We have reworked the caption of the figures.

Line 139 – HE and PSK – no transcription given

R= Thank you for your suggestion. We have included the definition.

Line 147 – maybe disactivated?

R= Corrected, thank you.

Discussion. The first sentence is a repetition of the following text, so it can be deleted.

R= Thank you for your suggestion. We have deleted the sentence.

Line 267. Authors says “lipidemic-lowering effect”, in the results it is clearly demonstrated only TG lowering, so maybe it will be better to change the phrase into TG lowering?

R= Corrected, thank you.

Line 295. It seems “It” is missing here. Also, the reference is needed in this sentence.

R= Corrected, and we have included the reference. Thank you.

Reviewer 4 Report (New Reviewer)

The authors in this research article determined “PPAR alpha activation by clofibrate alleviates ischemia/reperfusion injury in metabolic syndrome rats by decreasing cardiac inflammation and remodelling and by regulating the atrial natriuretic peptide compensatory response”. In the present research article, the authors have Natriuretic peptides (NPs) are cardiac hormones with numerous beneficial effects mainly mediated by a cell surface receptor named atrial natriuretic peptide receptor (ANPr). Although NPs are powerful clinical markers of cardiac failure, their role in I/R is still controversial. Peroxisome proliferator-activated receptor agonists exert cardiovascular therapeutic actions; however, their effect on the NPs signaling pathway has not been extensively studied. Our study provides important insight into the regulation of both ANP and ANPr in the hearts from MetS rats and their association to the inflammatory conditions caused by damage from I/R. Moreover, we showed that the pre-treatment with clofibrate was able to decrease the inflammatory response that in turn, decreases myocardial fibrosis, the expression of metalloproteases 2 and 9 and apoptosis. The treatment with clofibrate was also associated with the decrease of ANP and ANPr expression.

I can see very few articles in the present topic, which adds advantage for this study to be novel even though there are some flaws in the of the article. I would like to recommend some minor concerns to the authors to fulfil the scarce experimental evidence to prove the mechanism in the article.

v  In introduction, the authors have elaborately written the introduction, but details about need to be clearly PPAR alpha function in detail to be explained. Kindly check. https://doi.org/10.1016/j.phymed.2021.153648.

v  In materials and method some experiments are not clearly explained, please refer doi: 10.3389/fmolb.2022.1030534, https://doi.org/10.1186/s12929-022-00871-6 please correct.

v Why only one dose was chosen for the therapeutic study of clofibrate (Clo, 100mg/kg/day), kindly explain.

v Did the authors do pharmacokinetics, what was the concentration of clofibrate in the heart and plasma, kindly explain.

v The experiments pertaining the mechanism is not clear, discuss few justification for your novelty and mechanism.   

v  The authors need to show the final schematic conclusion diagram revealing a summarised mechanism for a layman understanding.

v  The authors need to concentrate in the requested changes as per review comments and check the manuscript for some grammatical errors and mistakes.

The available research information seems to be insufficient, and the authors need to address the above comments. Taking together to all this issue I recommend minor revision to the manuscript in present form.

Author Response

This manuscript is a resubmission of an earlier submission. The following is a list of the peer review reports and author responses from that submission.

Round 1

Reviewer 1 Report

In the manuscript “PPAR alpha activation by clofibrate alleviates ischemia/reperfusion injury in metabolic syndrome rats by decreasing cardiac inflammation and remodeling and by regulating the atrial natriuretic peptide compensatory response”, by Maria Sánchez Aguilar et al., A study explaining role of PPAR alpha activation by clofibrate inhibited IR injury in metabolic syndrome rats. I have the following concerns.

·     I have the following concerns.

1.    Authors have measured general characteristic and serum parameters in Mets-V and Mets-Clo without including IR group. It is interesting to know on various parameters post IR.

2.    Details of Blood pressure measurements is not given, in procedure it is mentioned that plethysmography was used for BP measurements. In cited reference (17), I could not find plethysmography word.

3.    Authors have pretreated with clofibrate in metabolic syndrome rats (most reported prophylactic action of it in IR injury). It is interesting to know the effect of clofibrate after ischemia/reperfusion injury in rats. Is there any effect of clofibrate on myocardial infarct volume?

4.    In some figure, there is not unclear band in WB.

5.    Justify selection of N in each experiment

6.    Data should be checked for normality before applying statistical test.

7.    For histology authors have used 3 rats which is insufficient to draw effective conclusions.

8.    N is not given for each figure legend.

Author Response

Dear Prof. Dr. Kornelia E. Jaquet,

Thank you for giving us the opportunity to submit a revised draft of the manuscript “PPAR alpha activation by clofibrate alleviates ischemia/reperfusion injury in metabolic syndrome rats by decreasing cardiac inflammation and remodeling and by regulating the atrial natriuretic peptide compensatory response” for publication in International Journal of Molecular Sciences. We appreciate the time and effort that you and the reviewers dedicated to providing feedback on our manuscript and we are grateful for the insightful comments and valuable improvements to our paper. We have incorporated most of the suggestions made by the reviewers. The changes done are in red within the manuscript. Please see below, in red, for a point-by-point response to the reviewer’ comments.

Reviewer 1

  1. Authors have measured general characteristic and serum parameters in Mets-V and Mets-Clo without including IR group. It is interesting to know on various parameters post IR.

R= We agree with the reviewer`s observation. The biochemical parameters presented in table 1 were determined in overnight fasting animals to prove that the administration of clofibrate diminished the levels of triglycerides and improved insulin sensitivity in MetS animals. Unfortunately, we were unable to obtain enough serum samples from animals subjected to I/R to be able to include them in Table 1.

  1. Details of Blood pressure measurements is not given, in procedure it is mentioned that plethysmography was used for BP measurements. In cited reference (17), I could not find plethysmography word.

R= Thank you for your observation. The tail‐cuff plethysmography technique is a common and non-invasive method used to determine physiological and pathological blood pressure in murine models. We replaced the term plethysmography by the full name in the methods section (4.1).

  1. Authors have pretreated with clofibrate in metabolic syndrome rats (most reported prophylactic action of it in IR injury). It is interesting to know the effect of clofibrate after ischemia/reperfusion injury in rats. Is there any effect of clofibrate on myocardial infarct volume?

R= PPAR agonists are drugs that are prescribed primarily to decrease triglyceride levels in clinical practice and several studies have demonstrated the association of the administration of fibrates with the decrease in myocardial failure. Therefore, we decided to evaluate the protective role of clofibrate in our rat model in the present study. At the present, we have not evaluated the effect of clofibrate after the myocardial injury in our experimental model and this could be an interesting study that must be approached with a different methodological strategy. The results of the proposed experiments are beyond the scope of the present paper and could be the object of another publication.

Regarding the effect of clofibrate on the infarct area we did not evaluate the clofibrate effect on myocardial infarct volume in the present paper. However, we suppose that clofibrate could have this effect since, we analyzed the ischemic size using the 2,3,5-triphenyl tetrazolium hydrochloride (TTC) staining in a previous report by our group, in which we demonstrated that fenofibrate treatment was able to attenuate myocardial damage evidenced by the reduction of the area at risk and of the ischemic area in MetS rats subjected to ischemic damage. (Sánchez-Aguilar M, Ibarra-Lara L, Del Valle-Mondragón L, Soria-Castro E, Torres-Narváez JC, Carreón-Torres E, Sánchez-Mendoza A, Rubio-Ruíz ME. Nonclassical Axis of the Renin-Angiotensin System and Neprilysin: Key Mediators That Underlie the Cardioprotective Effect of PPAR-Alpha Activation during Myocardial Ischemia in a Metabolic Syndrome Model. PPAR Res. 2020; 2020:8894525. doi: 10.1155/2020/8894525)

  1. In some figure, there is not unclear band in WB.

R= Thank you for your observation. In the revised version of our manuscript, we improved the quality of the images (figure 3).

  1. Justify selection of N in each experiment

R= We followed the 3 R principles (Replacement, Reduction and Refinement) to minimize the number of animals and respecting the most ethical use. Moreover, in previous reports we have demonstrated that under our experimental conditions, the use of 6 animals per group our renders our results reliable and reproducible. Specifically, for histology analysis, at least 4 fields (20X) of each animal (3 per group) were quantified, with a total of at least 12-24 determinations per condition. We added this information in methods section.

  1. Data should be checked for normality before applying statistical test.

R= Thank you for your observation. We repeated the statistical analysis using Graphpad software; we revised the data distribution and when the data were not normally distributed, we applied Kruskal Wallis tets. We specify this information in 4.12 methods section.

  1. For histology authors have used 3 rats which is insufficient to draw effective conclusions.

R= Thank you for your observation. Most of histological data reported in the literature do not inform on the n of animals used to obtain the images and only state the number of images analyzed which is usually between 6 to 12 images. In our study we are stating that we analyzed 4 to 8 images from the tissue that was obtained from 3 animals in each group.  We added this information in methods section.

  1. N is not given for each figure legend.

R= Thank you for your observation. We added this information in each figure legend.

Reviewer 2

This article has many problems, and the methodology was inadequate in several analyses. Therefore, this manuscript should not be published in this form as it is. Instead, I send criticisms and suggestions for future publication by the authors below.

Introduction

  1. There is no connection between the first and second paragraphs.

R= Thank you for your suggestion. We added the information about this issue and we citing the reference.

  1. Information on ischemia and reperfusion in the second paragraph is "loose." To improve their writing, the authors should ask themselves about the relationship between IR and heart failure.

R= Thank you for your suggestion. We have reworked introduction section according to reviewer suggestion.

  1. The third paragraph must start with a link to the second paragraph.

R= Thank you for your suggestion. We have reworked introduction section according to reviewer suggestion.

  1. Why do the authors use the term diabetic cardiomyopathy in line 77? Is the model used in the study a model of diabetes?

R= We agree with the reviewer`s observation. The rats used in the present study belonged to a MetS model and were not diabetic rats. We replaced the term “diabetic cardiomyopathy” for myocardial damage.

  1. Regarding paragraphs 87-89, are only these drugs used to treat metabolic syndrome? Is there a specific drug?

R= Since MetS is a multifactorial disease including different signs, drugs against each sign are often prescribed such as: biguanides, thiazolidinediones, ACE Inhibitors and Angiotensin II receptor blockers (ARBs); antiplatelet agents, metformin and lipid-lowering agents, such as statins and fibrates. Nevertheless, since fenofibrate has pleiotropic effects that favor cardioprotection, we decided to evaluate their effects on the expression of natriuretic peptides, inflammation, fibrosis and apoptosis.   

  1. What is the study's experimental question, hypothesis and objective? Again, this needs to be made clear in the introduction.

R= Thank you for your suggestion. We have reworked the introduction section to clarify.

 Results

  1. Why are the results of 4 groups shown in the table, and in figure 1, 6 groups are shown? In addition, authors should modify the format presented for a table.

R= We agree with the reviewer`s observation. The biochemical parameters presented in table 1 were determined in overnight fasting animals from to prove that the administration of clofibrate diminished the levels of triglycerides and improved insulin sensitivity in MetS animals. Unfortunately, we were unable to obtain enough serum samples from animals subjected to I/R to be able to include them in Table 1.

  1. I suggest presenting the graph data in another way. It's not didactic to keep turning your head to read the graphs. 9. The statistical difference symbols * and # as well as % and & are for the same group. Why? It needs to be clarified!

R= Thank you for your observation. As suggested by the reviewer, we replaced symbols for lowercase letters to indicate statistical differences in the graphs. We also made this modification in the figure legends.

  1. Why was the material used for HE and PSR staining not fixed as soon as it was collected? I don't understand the need for freezing. This influenced the quality of the material, which was not good. In addition, the methods section needs to inform what will be analyzed with HE and PSR and how it will be analyzed (detailed description). Furthermore, the methodology is not referenced.

R= Thank you for your observation. For the histological analysis, we used previously frozen tissue, since we had originally planned to perform the detection of markers by WB. However, since the presence of the proteins in the tissue is very low, we modified the experimental approach and decided to determine them by immunofluorescence in histological preparations. Freezing is a strategical technique that preserves tissues; moreover, there are previous studies that demonstrate that cardiac tissue resists a certain degree of freezing preserving its structural integrity and viability (Elami, et al.,  2008). It is important to state that the unfreezing process was performed in a gradual manner (Slow warming), before fixing the tissue for the HE and PSR staining (Marques et al., 2019 ; Rivas Leonel et al., 2019 ). Furthermore, tissues were cryo-protected with 30% sucrose as described in the material and methods section. Moreover, the treatment of the tissues was the same for all of the experimental groups and alterations from the procedure would be equivalent for all of them, avoiding the possibility that differences could come from the procedure instead of from the treatments being tested.

We have included this information in the methods section and we are citing the corresponding references.

  1. Figure 2A. The quality of the histologies is not good. The authors demonstrate the quantification of inflammatory cells. What was the parameter of this analysis? How did the authors distinguish fibroblasts, myocytes and inflammatory cells?

R= In the revised version of our manuscript, we improved the quality of the image (figure 2). The number of infiltrated cells within the heart muscle fibers was performed in 12 different images of each tissue. The presence of infiltrates is considered as an indicator of an inflammatory process. In our study, this was corroborated by the quantification of proinflammatory cytokines in the heart tissue.  It is reported in the literature that the most common infiltrates comes from neutrophils and/or macrophages. It would be important to further address this issue performing neutrophil or MAC-2 stainings or using specific markers for each cell type such as CD5, CD20, CD68, but unfortunately, this was beyond the aim of the present study. We added a paragraph in discussion section.

  1. Why is there a big difference between PSR photos in fluorescence and white light? Is fluorescence a suitable method for this analysis? Also, the graphics need to have unity. For example, what does a signal of 200 represent for collagen? Finally, the authors need to see how these data are described in the literature, which is being expressed here and needs to be corrected.

R= Results obtained for PSR by light microscopy and fluorescence microscopy are equivalent. The analysis made by fluorescence is not the most frequently used method for PSR; however, we were showing it since it is a sensitive and reproducible method that allows to have an alternative to identify the regions of interest for the quantification of collagen in tissue slices (Wegner,et al., 2017 ). We decided to modify and improve Figure 2 showing only the values of PSR detection obtained by fluorescence. To evaluate the extension of myocardiac fibrosis as deposits of collagen, we are showing the values as percentages of the collagen volume fraction (CVF) as previously reported by other authors (Lu et al., 2019 ).  This corresponds to the analysis of the area of collagen with respect to the total area of the field of the images acquired at 20X using the Image-Pro Premier 9 (Media Cybernetics). We added the bibliographic references corresponding to this issue.

  1. Figure 3, same confusing repetition of symbols in the differences. I want the entire membranes to be presented because, as in the figure and supplementary material, it is impossible to see if the marking is adequate and correct. Antibodies to MMP-9 have a major specificity problem.

R= In the revised version of our manuscript, we improved the quality of the image (figure 3). However, at the request of the reviewer, we present to you the images that we obtained of the MMP-2 and MMP-9 expression. It is worth mentioning that the ten antibodies used to perform western blot analysis were monoclonal antibodies and had been previously used in several publications by other authors.

  1. Figures 4 and 5, Tissue quality is not good. Again tissue freezing without previous fixation or use of cryoprotection liquid was harmful. The authors do not inform in the methods how the expression was quantified. Was the fluorescence intensity quantified?

R= The section of the heart corresponding to the left ventricle was separated at the moment of the extraction of the heart. It was carefully manipulated to eliminate excess liquid and it was immediately placed at -70°C. For the histological analysis, we used previously frozen tissue, since we had originally planned to perform the detection of markers by WB. However, it was not possible to detect ANP in the tissue by WB analysis since the presence of the proteins in the tissue is very low. Hence, we modified the experimental approach and decided to determine them by immunofluorescence in histological preparations. It is important to state that the unfreezing process was performed in a gradual manner (Slow warming),passing tissues from  -70°C  to -20°C, then to 4°C and finally to room temperature before fixing the tissue with paraformaldehyde (PFA) 4%. After the fixation period, the tissue was carefully washed and cryo-protected with 30% sucrose. Moreover, since the tissues from all of the experimental groups were equally treated, the possible alterations from the procedure would have been the same without affecting the effects caused by the experimental procedure. We added the detailed information in the methods section. The fluorescence intensity was reported as IOD, that corresponds to the intensity per unit of area (lum/pix^2). We have included this information in the methods section and we are citing the corresponding references.

  1. Were anti-ANP and anti-ANPR immunofluorescence performed with antibodies incubated for 72h? Please justify this time and how specific and unspecific binding was verified.

R= We chose the time of the incubation for the primary antibodies after analyzing other previous studies using this time of incubation (72 h a 4°C) with good results (Butler et al., 2019). This experimental approach was chosen after having had negative results for the detection of ANP by WB. Regarding specificity, the antibodies, they were monoclonal antibodies and we used them according to the information provided by the supplier (data sheet ab225844 and data sheet ab14356 ). The non- specific binding was verified in negative controls that were incubated with the blocking solution (BSA 3%/0.1 triton X-100, in PBS) without the primary antibody. All precautions were taken including the incubation in a Glycine 0.1M/PBS pH 7.4 solution for 2 min to eliminate the possible quenching generated by the aldehydes in the fixing solution. We also performed the antigenic recuperation with Tris (0.5M)/EDTA (0.1M), pH 9 solution at 95°C for 10 min. We have included this information in the methods section and we are citing the corresponding reference.

  1. Table 2. The results must be precise. If a statistical test was used and there was no difference, then obviously, there is no difference! Modify table 2 to table format, as this is a frame.

R= We agree with the reviewer. In the revised version of the manuscript, we state that “There is no statistical difference among the groups for NT-proBNP concentrations” in results and discussion sections. We replaced Table 2 for a figure (Figure 6).

  1. Figure 6. How were samples collected for this analysis?

R= Thank you for your observation. We added the complete information for immune colloidal gold technique in 4.11 methods section.

  1. Figure 7. How was the apoptosis outcome quantified? It should be mentioned in the methodology. Also, why did the authors use a previously stained slide? In addition to the visualization not being good, does this not interfere with the apoptosis test?

 R= Thank you for your observation. Apoptosis was quantified in images in which the nuclei were marked with DAPI. The total number of blue nuclei per field and the number of nuclei positive for TUNEL (green) were quantified and the percentage of cells in apoptosis was calculated in at least 12 fields from each condition, The double marking (PNA/TUNEL) was used according to the indications of the supplier that states that this is possible (Roche-11684795910 ) and according to other reported studies (Oberhaus  2003 ; Christine:, 2006 ; Kyrylkova et al 2012  ). Even if there was an interference it should have been equivalent in all of the slides, and therefore, the differences should be attributed to the treatments.  We consider that the visualization of the TUNEL+ cells was clearly distinguishable. We added the complete information for the technique in 4.9 methods section and we are citing the corresponding references.

Discussion

  1. Based on the results obtained, the authors could not answer the experimental question clearly and objectively. Furthermore, the discussion is vague and extensive. Many points are not adequately discussed, such as the experimental model, the drug used, the role of natriuretic peptides, the role of MMPs in ischemia and reperfusion (there is a lot of data in the literature), as well as the relationship of inflammation, natriuretic peptides, MMPs, remodeling and apoptosis.

R= The manuscript was revised, and we have reworked some section of discussion according to reviewer suggestion.

Materials and methods

  1. Reference methodology

R= Thank you for your observation. We added several references for the 4.6, 4.7 and 4.8 methods section and included them in the reference list.

  1. Why was treatment chosen before ischemia and not after the procedure?

R= In clinical practice, PPAR agonists are drugs that are prescribed primarily to decrease triglyceride levels, and several studies have demonstrated the association of the administration of fibrates with a decrease in myocardial failure. That is why, we decided to evaluate the protective role of clofibrate in our rat model in the present study. At present, in we have not evaluated the effect of clofibrate after myocardial injury in our experimental model. This could be an interesting study needing for a different methodological strategy. The results of these experiments are beyond the scope of the present paper and could be the object of another publication.

Best regards,

María Esther Rubio Ruiz PhD

Department of Physiology,

Instituto Nacional de Cardiología

“Ignacio Chávez”

esther_rubio_ruiz@yahoo.com

Reviewer 2 Report

This article has many problems, and the methodology was inadequate in several analyses. Therefore, this manuscript should not be published in this form as it is. Instead, I send criticisms and suggestions for future publication by the authors below.

Introduction

1. There is no connection between the first and second paragraphs.

2. Information on ischemia and reperfusion in the second paragraph is "loose." To improve their writing, the authors should ask themselves about the relationship between IR and heart failure.

3. The third paragraph must start with a link to the second paragraph.

4. Why do the authors use the term diabetic cardiomyopathy in line 77? Is the model used in the study a model of diabetes?

5. Regarding paragraphs 87-89, are only these drugs used to treat metabolic syndrome? Is there a specific drug?

6. What is the study's experimental question, hypothesis and objective? Again, this needs to be made clear in the introduction.

Results

7. Why are the results of 4 groups shown in the table, and in figure 1, 6 groups are shown? In addition, authors should modify the format presented for a table.

8. I suggest presenting the graph data in another way. It's not didactic to keep turning your head to read the graphs.

9. The statistical difference symbols * and # as well as % and & are for the same group. Why? It needs to be clarified!

10. Why was the material used for HE and PSR staining not fixed as soon as it was collected? I don't understand the need for freezing. This influenced the quality of the material, which was not good. In addition, the methods section needs to inform what will be analyzed with HE and PSR and how it will be analyzed (detailed description). Furthermore, the methodology is not referenced.

11. Figure 2A. The quality of the histologies is not good. The authors demonstrate the quantification of inflammatory cells. What was the parameter of this analysis? How did the authors distinguish fibroblasts, myocytes and inflammatory cells?

12. Why is there a big difference between PSR photos in fluorescence and white light? Is fluorescence a suitable method for this analysis? Also, the graphics need to have unity. For example, what does a signal of 200 represent for collagen? Finally, the authors need to see how these data are described in the literature, which is being expressed here and needs to be corrected.

13. Figure 3, same confusing repetition of symbols in the differences. I want the entire membranes to be presented because, as in the figure and supplementary material, it is impossible to see if the marking is adequate and correct. Antibodies to MMP-9 have a major specificity problem.

14. Figures 4 and 5, Tissue quality is not good. Again tissue freezing without previous fixation or use of cryoprotection liquid was harmful. The authors do not inform in the methods how the expression was quantified. Was the fluorescence intensity quantified?

15. Were anti-ANP and anti-ANPR immunofluorescence performed with antibodies incubated for 72h? Please justify this time and how specific and unspecific binding was verified.

16. Table 2. The results must be precise. If a statistical test was used and there was no difference, then obviously, there is no difference! Modify table 2 to table format, as this is a frame.

17. Figure 6. How were samples collected for this analysis?

18. Figure 7. How was the apoptosis outcome quantified? It should be mentioned in the methodology. Also, why did the authors use a previously stained slide? In addition to the visualization not being good, does this not interfere with the apoptosis test?

Discussion

19. Based on the results obtained, the authors could not answer the experimental question clearly and objectively. Furthermore, the discussion is vague and extensive. Many points are not adequately discussed, such as the experimental model, the drug used, the role of natriuretic peptides, the role of MMPs in ischemia and reperfusion (there is a lot of data in the literature), as well as the relationship of inflammation, natriuretic peptides, MMPs, remodeling and apoptosis.

Materials and methods

20. Reference methodology

21. Why was treatment chosen before ischemia and not after the procedure?

Round 2

Reviewer 1 Report

In revised version, I have no further comments except: Please provide unedited blot for WB.

Reviewer 2 Report

The manuscript could have been written better. In addition, it has serious methodological problems and problems with presenting results and discussion. Therefore, I do not recommend the publication of the manuscript, and I indicate below the reasons for the refusal:

  1. Improve the title, making it more concise and similar to the objective.
  2. Rewrite the introduction trying to make a relationship between all the points to be addressed in the manuscript.
  3. Low experimental “n” requires a second round of experiments to confirm the observed effects.
  4. Add the missing groups in table 1 (the justification given cannot be accepted).
  5. Modify the form from frame to table (this problem persists).
  6. Better explain the experimental model and the importance of pre-treatment (the authors could not or do not know how to justify it).
  7. The authors should better explain the work developed because the way it is, it looks like two independent studies that were joined.
  8. The methodology for histology and immunofluorescence needed to be adequate, which impaired the quality of the results, in addition to using photos at higher magnification.
  9. The authors still used photos that are not representative of the graphics;
  10. Western blotting performed by the authors are not recommended for publication, as the membranes sent showed several marked bands, which demonstrates low selectivity and experimental problems;
  11. The graphs are still presented in a non-didactic way. I suggest using acronyms horizontally to facilitate reading and using colors in the graphics;
  12. Regarding statistics, the authors considered p < 0.05 statistically significant, so they should adopt only this significance and remove symbols with other significances in the figures and captions. This would make the charts much easier to read;
  13. The overlapping of the images in the immuno shows that they are not on the same plane;
  14. I want the authors to explain how the increase in collagen, ANP and ANPR expression occurs in a few hours and the relationship between this and the absence of difference in the result with NT-pro-BNP;
  15. The justification given by the authors for the use of immuno slides for the tunnel experiment was not convincing;
  16. The worst part of the manuscript is the discussion. First, the authors should draw attention to the data found in the study and then discuss point by point with studies in the literature. Furthermore, the discussion is a litany difficult to read and tiresome for its length;
  17. The conclusion should be more direct and succinct;